# Influence of Machining Parameters on Cutting and Chip-Formation Process during Cortical Bone Orthogonal Machining

**DOI:** 10.3390/ma15186414

**Published:** 2022-09-15

**Authors:** Paweł Zawadzki, Rafał Talar, Adam Patalas, Stanisław Legutko

**Affiliations:** Faculty of Mechanical Engineering, Poznan University of Technology, Maria Sklodowska-Curie Square 5, 60-965 Poznan, Poland

**Keywords:** abrasive machining, orthogonal cutting, cortical tissue, cutting force, coefficient of friction, precise shaping

## Abstract

Cortical bone machining is commonly used in craniofacial surgery. The shaping of bone surfaces requires a precise determination of the process’s complexity due to the cutting tool’s defined or undefined geometry. Therefore, research was carried out to assess the impact of the rake angle (*γ*), clearance angle and depth of cut (*d*) on the cortical bone machining process. Analysis was carried out based on the orthogonal cutting in three directions. The cutting tool shape was simplified, and the cutting forces and the chip-formation process were monitored. The highest values of the resultant cutting force and shear force were recorded for *γ* < 0. The specific cutting force decreases with the increase of *d*. Cutting in the transverse direction is characterized by the highest values of resultant cutting force and shear force. The coefficient of friction depends primarily on the *d* and takes a constant value or increases with the increase of *γ*. The tests showed that the chips are formed in the entire range of *d* ≥ 0.5 µm and create regular shapes for *d* ≥ 10 µm. The research novelty confirms that even negative cutting angles guarantee controlled cutting and can find wider application in surgical procedures.

## 1. Introduction

Orthopedic surgery includes various invasive machining methods applied to bones and joints. The primary orthopedic surgery procedures include hip arthroplasty, knee arthroplasty [1], limb fracture surgery and other joint replacement techniques [2]. The number of worldwide orthopedic surgeries amounted to approximately 22.3 million, and it is one of the fastest-growing categories of medical procedures [3]. One of the primary methods used during orthopedic procedures is cartilage and bone tissue surface machining based on drilling, cutting and milling [4].

Many scientists have carried out analyses of bone tissue machining. Lee et al. [5] researched bone tissue drilling and conducted experimental and simulation analyses of the thrust and torsion mechanical force model during bone drilling. The thrust forces and torque differences were observed between bone samples from different animals. Another procedure is tissue cutting, on which James et al. [6] conducted research, presenting an analytical model for predicting bone sawing forces, which was compared with experimental tests. Moreover, Yaping et al. [7] prepared the simulator for bone tissue cutting and conducted extensive experimental and simulation tests, showing the value of the forces accompanying the cutting at 4–7 N. Another method is milling, which is discussed in more detail in Liao et al. [8]. It has been shown that the cutting force for a single cutting-edge tool depends on the thickness of the chip. It also showed thermal necrosis penetration depth with increased cutting speed. Despite the growth in cutting speed, the temperature remains the same due to the rise in global heat flux. Similar results were shown by Chen et al. [9], in which the resultant milling force was obtained at the level of 2 to 14 N.

Unfortunately, all types of mechanical impact involve thermal energy emission, tissue destruction and extension of recovery time [10]. Moreover, the popularity of standardized surgical procedures causes no changes in the processing technology and the availability of specialized surgical equipment. It should be noted that bone tissue is a complex material that includes microstructure elements (osteons with a diameter of 100–300 μm) and sub-microstructures (bone lamellae with sizes 3–7 μm) [11]. The specific distribution of osteons results in material anisotropy. Cortical bone is a quasi-brittle living tissue, and these properties result in the easy propagation of cracks in specific structure directions. Failure to control fractures can damage the blood vessels and nerve endings. 

Therefore, to protect the patient’s health, special attention should be paid to the mechanism of cutting the bone tissue. In the following studies, the focus was paid to the cortical bone tissue due to the apparent variability in the mechanical properties of various cartilage bone tissues. It is built of concentrically arranged bone plates, forming structural and functional units of the osteon bone. The lamellae are shaped like cylinders inserted into each other and contain a centrally located osteon channel. Many scientists compare its structure to the composite structure. Therefore, it is characterized by variable properties depending on the direction of measurement of the parameter. 

Grinding–machining of bones is widely used in cranial and craniofacial surgery. Yahui et al. [12] presented a model and experimental studies to minimize craniotomy. The experimental results confirmed the model predictions in which the grinding force increased with the feed speed and depth of grinding. Babbar et al. [13] presented in vivo research on the evaluation of machining forces, torque and bone quality during skull bone grinding. The three process parameters, namely rotational speed, feed rate and depth of cut, were examined at three levels for the tangential force, contact force and torque generated during grinding. Similar studies were presented by Zhang et al. [14], focusing on the temperature emission during grinding. The high spindle speed would cause the temperature to rise; however, it significantly reduces the grinding force. A reduced cut and feed rate depth at a given spindle speed helps reduce the grinding temperature and force.

The research presented above focuses on analyzing complex shearing processes involving grain assemblies. They provide complex information about the processes taking place on a macroscale, and it would be advisable to analyze the abrasion process on a microscale based on the basic orthogonal cutting model. Orthogonal cutting is one of the fundamental processes of bone tissue surgery, although its characteristics require constant research. The most comprehensive research in this area has been carried out by Liao et al. [15] and Bai et al. [16]. Liao et al. [15] analyzed the chip-formation mechanism using one tool with rake angle *γ* = 8° and clearance angle *α* = 8° for the depth of cut *d* = 5 to 200 μm. As a result, three models of bone fracture were based, and surface morphology and cutting force were described. Bai et al. [16] performed analyses using the tool with rake angle *γ* = 10° and clearance angle *α* = 7° for the depth of cut *d* = 10, 50 and 150 μm. As a result, the chip-formation process, cutting forces and surface morphology were characterized. Orthogonal cutting studies have been carried out to a reduced extent by Jacobs et al. [17] in pioneering analyses describing the complexity of bone tissue in three directions. Wiggins and Malkin [18] focused on the description of chip formation, suggesting that discontinuous chip occurs by a series of discrete fractures. Similar analyses were performed by Sugita et al. [19] and Plaskos et al. [20], indicating that the cutting depth has the most critical influence on the shape of the produced chips. In most cases presented above, a single cutting tool was used.

Therefore, in the following study, orthogonal cutting analyses were carried out on a broad spectrum of cutting tool geometries to map the undefined cutting geometry of abrasive grains. In addition, a wide range of cutting depths with a tool with a defined geometry was introduced due to the variable value of the actual cutting depth of the abrasive grains. The above variables—rake angle, clearance angle and depth of cut—were intended to reflect the complexity of the cutting process with the use of abrasive tools as faithfully as possible. Particular attention was paid to microstructural features and anisotropy of the cortical bone tissue. Based on the obtained data, the values of the forces during the cutting process were determined, and the specific cutting force of the cortical bone tissue was determined. The chip-formation process and coefficient of friction, depending on the orientation of the bone tissue, were also characterized.

## 2. Materials and Methods

### 2.1. Plan of the Experiment

The study aimed to analyze the complexity of abrasive machining. For this purpose, the orthogonal cutting of cortical bone tissue was used. The shape of the abrasive grain was simplified to a simple cutting tool. A wide range of input parameters (such as tool geometry and cutting depth) was used to approximate the complexity of the processes taking place during cutting. Cutting forces and acoustic emission were measured during the experiments, and the resulting chips were photographed (see Figure 1). 

### 2.2. Bone Characteristics

The cortical bone is an essential part of the human skeletal system and is characterized by about 5–10% porosity [11]. Bone is a heterogeneous composite material with constituents including hydroxyapatite mineral (Ca10 (PO4) 6 (OH) 2) [21] and water [22,23]. The basic building blocks of cortical bone tissue are osteons, taking the shape of elliptical cylinders with a diameter of 100–300 µm and a length of 3–5 mm [24,25,26]. It surrounds the Haversian canals and is embedded in the interstitial tissue, the remnant of old osteons. The osteons are separated from the interstitial bone tissue by a thin layer of amorphous substance deficient in collagen called a cement line (0.5 to 1 μm). Each osteon is made of concentric lamellae (1 to 5 μm thick), among which the bone cells reside inside ellipsoidal spaces called lacuna (10 to 50 μm).

Cortical bone tissue differs in the calcium, phosphorus, water content, density and degree of mineralization from cancellous bone tissue [27,28]. The elastic modulus in the longitudinal direction in long bone (17.4 GPa for human bone and 20.4 GPa for bovine bone) is more significant than that in the transverse (9.6 GPa for human bone and 11.7 GPa for bovine bone) [29]. It was assumed that the bone tissue should be cut in three directions. Additionally, due to the similar properties of human and bovine bone tissue, use identity can be assumed. The following properties of bone tissue, presented in Table 1, were adopted for the research. Isotropic properties were adopted following the assumptions of Cowin and Sadegh [30]. As indicated by the results of studies by Abdel-Wahab et al. [31] and Sugita et al. [32], interstitial matrix has a slightly higher Young’s modulus (22.8 GPa) than cortical bone. The lower Young and strength moduli characterize the cemented line. 

### 2.3. Bone Specimens Preparation

Fresh bovine femurs were obtained from the local slaughterhouse and maintained in Ringer’s fluid. Proximal and distal ends of the femoral bone were removed, leaving the central section. The mechanical properties of bovine femoral bone are similar to human cortical bone [33]. Animal-derived bones are used in most experimental studies [8,12,13,14,15,16]. The fragments were cut from the central part of the diaphysis. The size of the prepared samples was 20 mm × 4 mm × 10 mm, including width of 4 mm. Due to the anisotropic structure of cortical bone, it was necessary to construct specimens with three independent orientations where the direction of the osteons was a reference [34] (see Figure 2). A fresh femur was cut in the radial, axial and circumferential directions to take bone samples across parallel and transverse directions. The samples were polished with 320 grit sandpaper, dipped in a Ringer’s fluid and stored in a refrigerator at 0 °C (after tests at −25 °C). The literature shows no statistically significant differences compared to the mechanical properties of the cortical bone frozen to fresh [35,36]. Bryan et al. [37] concluded that frozen bone samples could be used in research on bone mechanical properties.

### 2.4. Cutting Tool Geometry

In order to carry out parameterized analyses, it was assumed that the abrasive tool consists of an abrasive grain assembly with undefined geometry, although it is possible to analyze a wide range of cutting tools with defined geometry that will allow reproducing the actual state (see Figure 3). The author designed the high-speed steel cutting tools, and it was produced by Avanti-Tools Sp. Zoo. (Poznan, Poland). The tool’s rake angles *γ* and the clearance angles *α* values were presented in Table 2. An example of the cutting tools is shown in Figure 4. The width of the tool cutting edge was defined as 10 mm. In experiments with orthogonal cutting, the width of the cutting edge should be much larger than the width of the workpiece. The tools were mounted in a specialized holder made of structural steel.

It should be noted that such a wide range of tool geometry (27 cutting tools were prepared) has not been analyzed yet (see Table 2).

### 2.5. Experiment Parameters

The following input parameters were used, as shown in Table 2. The wide range of cutting tool geometry reflected the geometry of the abrasive grains. Cutting depths were based on variable values of cutting depth depending on the conditions prevailing during cutting with an abrasive grain assembly. The range of 0.5 to 50 µm covers lamellar thickness, and higher values of 50 to 200 µm cover the size of the osteon diameter [38]. A constant cutting speed of *v_c_* = 30 mm/min was established to minimize the impact of thermal effects on the cutting process. Detailed test parameters and the configuration method are presented in Table 2.

### 2.6. Experimental Setup

The measuring system presented in Figure 5 was developed to carry out the full scope of the study of the orthogonal cutting process of cortical bone tissue. The tool movement during orthogonal cutting was performed using UMT Bruker tribotester (Billerica, MA, USA) drives equipped with a 3-axis motion system and high-resolution stepper motors drives. The DFM-20 two-axis force sensor measured the force with a range of 0.05 to 235 N. The following features characterize it: measuring resolution of 0.01 N, non-linearity of 0.02% and sampling frequency of 1000 Hz. It provides precise measurement of force, position in three axes and *AE* generated during cutting.

The Motic optical microscope (Hongkong, China) with a microscope camera ensuring image registration was used to observe the cutting process. A camera with a maximum resolution of 2048 × 1536 pixels was used. Pictures were taken during chip formation. In addition, microscopic measurements of the chips were carried out on dedicated software (Motic Image Plus 2.0, Motic, Hongkong, China). 

### 2.7. Determination of Cutting Forces

As a result of the measurements, the tangential force *F_c_* and the contact force *F_t_* were obtained. In further stages, analytical calculations were performed to determine the resultant cutting force *R* and shear force *S*. The analytical calculations were implemented on the improved cutting model based on Merchant [39], across the cutting model, considering the tool edge radius effect presented by Yang et al. [40]. Since the adopted depths of cut are small (0.5 to 200 µm), the cutting edge radius *r* can significantly affect the process. The measured corner radius of the tool was *r* = 104 ± 12 μm. 

In Figure 6A, the resultant cutting force *R* is the sum of the shear force *S* and the ploughing force *P*. According to the analysis of the Merchant model, the shear force *S* of the modified model can be divided into forces along the shear plane *F_s_* and across to the shear plane *F_n_*, as well as forces along cutting direction *F_c_* and across to cutting direction *F_t_*. At the chip contact area, the shear force can be distributed between the frictional force *F_f_* along the chip contact area and the force *N* across to the chip contact area. As with the shear force, the ploughing force can also be analyzed on the shear plane and chip contact area. Assuming the shear angle as *θ* and the rake angle as *α*:(1)$$\left\{\begin{array}{c} F_{s}=F_{c}cos\theta -F_{t}sin\theta \\ F_{n}=F_{c}sin\theta +F_{t}cos\theta \end{array}\right.$$
(2)$$\left\{\begin{array}{c} F=F_{c}\sin\gamma +F_{t}cos\gamma \\ N=F_{c}\cos\gamma -F_{t}sin\gamma \end{array}\right.$$

The shear angle *θ* was found by the following equation:(3)$$tan\theta =\frac{d_{\Delta }*\cos\gamma }{1-d_{1\;}*\sin\gamma }$$
where *d*_Δ_ is the ratio of the depth of cut-to-chip thickness *d*_1_. Chip thicknesses were assessed experimentally based on microscopic measurements. Based on the above information, *μ_tc_* friction coefficient of the tool–chip contact surface can be determined by the equation:(4)$$\mu_{tc}=\frac{F_{f}}{N}=\frac{F_{c}\sin\gamma +F_{t}\cos\gamma }{F_{c}\cos\gamma -F_{t}\sin\gamma }$$

Assuming that the contact surface of chips with the tool has the same friction coefficient value as in the tool’s contact zone with the processed material, it can be assumed that during the cutting process, *μ_tc_ = μ_tw_ = COF*. The ploughing force value can be reduced to the following equation:(5)$$P=\sqrt{F_{P}^{2}+N_{P}^{2}}$$
where
(6)$$\left\{\begin{array}{c} F_{p}=F_{f}\\ N_{p}=N \end{array}\right.$$
since
(7)$$\mu_{tc}=\frac{F_{f}}{N}=\mu_{tw}=\frac{F_{p}}{N_{p}}$$
that is
(8)$$P=\sqrt{F_{f}^{2}+N^{2}}$$
where
(9)$$F_{f}=F=F_{c}\;\sin\gamma +F_{t}\cos\gamma$$

According to the above assumptions, the following should be adopted:(10)$$\left\{\begin{array}{c} N_{p}=P_{t}=N\\ R_{t}=F_{t}+P_{t} \end{array}\right.$$

As a result, the shear force *S* can be determined:(11)$$S=F_{t}*\cos\gamma -F_{n}*\sin\gamma$$

Continuing to simplify:(12)$$\left\{\begin{array}{c} F_{p}=P_{c}=F_{f}\\ R_{c}=F_{c}+P_{c} \end{array}\right.$$

The *R* resultant cutting force value was obtained:(13)$$R=\sqrt{R_{t}^{2}}+R_{c}^{2}$$

Experimental measurement results also allowed to determine the direct value of the specific cutting force *k_c_* by the equation:(14)$$k_{c}=\frac{F_{c}}{A}=\frac{F_{c}}{w_{1}*w_{2}}=\frac{F_{c}}{w_{1}*\frac{d}{\cos\gamma }}=\frac{F_{c}*\cos\gamma }{w_{1}*d}$$
based on Figure 5B.

## 3. Results and Discussion

The cutting directions interpreted in the following points are shown in Figure 2.

### 3.1. Resultant Cutting Force

#### 3.1.1. Across Direction

The course of changes in the resultant value of the cutting force *R* should be divided into group A for *γ* ≤ 0° and B *γ* > 0° (see Figure 7A). For group A, the *R* values, regardless of the clearance angle *α*, are significantly greater than for group B. In all cases of *α*, a linear decreasing trend of the *R* value in the direction of *γ*+ with correlation coefficients *ρ* > 0.9 was visible. However, it has been observed that below *d* < 10 μm, the potential values of the resultant cutting force *R* are similar. In group B (*γ* > 0°), the characteristics of the curves are more irregular than in group A (*γ* ≤ 0°) and are characterized by high differences between rake angles *γ*. In group A, the values change predictably. Considering the clearance angle value *α*, the best correlation between the waveform and the *R* values is *α* = 5°. *R_max_* for individual angles of application *α* were: *R_max_* = 335 MPa (*α* = 5°), 411 MPa (*α* = 10°) and 352 MPa (*α* = 15°).

The increase in clearance angle potentially affected the regularity of changes, especially in the range of negative cutting angles *γ* < 0°. It can be concluded that increasing the *α* results in a change in the stress distribution in the workpiece space, which is essential for *γ* < 0° due to the compressive nature of cutting. The minimum value of the resultant cutting force *R_min_* for *γ* = −40° was for angles *α* = 5°, 10° and 15°, respectively 118 MPa, 79 MPa and 42 MPa. 

#### 3.1.2. Parallel Direction

Changes in the resultant cutting force *R* in the parallel direction duplicate the changes for the transverse direction (see Figure 7B). *R_max_* for individual angles of application *α* were: *R_max_* = 333 MPa (*α* = 5°), 363 MPa (*α* = 10°) and 439 MPa (*α* = 15°). The resultant cutting force *R* values for *α* = 15° are more significant than transverse and across curves. The tendency of the division into groups A and B is noticeable. Group A is characterized by a linear change with a correlation coefficient value of *ρ* > 0.98, similar to group B, but the coefficient decreases to *ρ* = 0.86 due to the difference of the high values between groups A and B. For *d* < 10, there is a noticeable tendency to maintain similar values, especially in the clearance angle *α* = 5° and 10 °. The minimum value of the resultant cutting force *R_min_* was obtained for *γ* = −40°, angles *α* = 5°, 10° and 15°, respectively 52 MPa, 24 MPa and 82 MPa. 

#### 3.1.3. Transverse Direction

The most significant discrepancy characterizes the movement towards the transverse in the resultant cutting force and its correlation coefficient of the results (*ρ* < 90) (see Figure 7C). Especially for *d* > 50 μm, the curves are irregular. However, it indicates a decreasing trend in the *R* value with a positive increase in rake angle *α* (see Figure 7D). It is characteristic that the linear course of the resultant cutting force is maintained for all cutting angles at *d* < 10 μm. For greater depths, the irregularity of the process increases, maintaining an upward trend. *R_max_* for individual angles of application *α* were: *R_max_* = 459 MPa (*α* = 5°), 377 MPa (*α* = 10°) and 285 MPa (*α* = 15°). The minimum value of the resultant cutting force *R_min_* was obtained for different rake angles and clearance angles *α* = 5°, 10° and 15°, respectively 14 MPa, 26 MPa and 17 MPa. 

The general values of the resultant cutting force *R* take the lowest values concerning the across and parallel directions (see Figure 8). The most significant irregularity of the courses characterizes the transverse direction and the achievement of the highest values of R. Increasing *R* indicates an increase in shear resistance, and the intensification of fluctuations indicates an increase in the propagation of cracks in the forming chip. The signal fluctuations increase for *d* > 50 μm, which is confirmed by Bai et al. [16].

The obtained results are confirmed by the research conducted by Liao et al. [15], which showed a clear tendency for the cutting force to increase with increased cutting depth. In addition, according to Liao et al. [15], a cutting process in the transverse direction results in the highest cutting force values. The results of Bai et al. [16} also confirm the current results, which indicate that the processes are regular at *d* < 50 μm, and then numerous signal fluctuations with an increase in *d*, resulting from an increase in pushing force and the formation of a non-uniform chip that resulted in the decrease of the separation force.

Similar to Sui et al. [41], a tendency was found for the *F_c_* value to increase for increasing y, and hence *R* with an increase in *γ*. Sui et al. [41] argue that this is due to the rake angle’s reduced contact length between the tool and the chip, and the shortened contact length reduces the friction force. Additionally, according to the observations, tools with *γ* < 0° tended to slide over the bone tissue’s surface while denting it perpendicularly to the direction of movement, increasing the specific cutting force. Tissue cutting and chip formation were limited. Increasing the rake angle resulted in a tendency for the blade to nest in the material, resulting in cutting regularity. 

Similarly to the studies by Sui et al. [41], there is a noticeable tendency for the intensification of changes in the *F_c_* value with the change of the depth of cut. Sui et al. [41] obtained *F_c_*
_=_ 35, 66 and 87 N for *d* = 20, 50 and 80 µm, for *γ* = −10° and across direction, respectively. The author obtained the following results: *F_c_* = 36, 57 and 89 N; and *R =* 65, 94 and 144 MPa. There is a noticeable correlation between the results. It should be noted that the author has researched a wider range of *d* and *γ*.

### 3.2. Shear Force

#### 3.2.1. Across Direction

The value of the cutting force *S* presents a similar characteristic to the resultant force *R* (see Figure 9A). Groups A and B, depending on the value of rake angle *γ*, can also be distinguished. *S_max_* for individual angles of application *α* were: *S_max_* = 212 MPa (*α* = 5°), 226 MPa (*α* = 10°) and 235 MPa (*α* = 15°). Figure 8 presents the course of *S* with a decrease in *γ* in a positive direction. It is worth noting that for *α* = 40° and *d* = 0.5−10 μm, and *α* = 30° and *d* = 0.5−10 μm for all angles of application, negative values of shear force were obtained, which indicates the pushing out of the tool of the material space. This confirms the observations regarding *R*, in which it is indicated that the value of *R* increased with the decrease in *γ* due to the reduction of the penetration of the tool into the tissue in favor of sliding, friction and crushing of the material.

#### 3.2.2. Parallel Direction

*S_max_* for individual angles of application *α* was: *S_max_* = 209 MPa (*α* = 5°), 205 MPa (*α* = 10°) and 266 MPa (*α* = 15°) (see Figure 9B). It is worth noting that for *α* = 40° and *d* = 0.5–25 μm, and *α* = 30° and *d* = 0.5−10 μm for all angles of application, negative values of shear force were also obtained.

#### 3.2.3. Transverse Direction

*Smax* for each angle of application *α* was: *S_max_* = 314 MPa (*α* = 5°), 192 MPa (*α* = 10°) and 192 MPa (*α* = 15°) (see Figure 9C). Curves tend to have a linear decrease in the *S* value with the change of rake angle to positive. It was observed that for α = 40° and d = 0.5−25 μm, and α = 30° and d = 0.5−10 μm, negative values of shear force were obtained—for all rake agles. The results of *S* measurements show a clear tendency for the value to increase with the increase of *d* and then decrease with the change of *γ* towards the positive direction (see Figure 9D). According to the work of Jacobs et al. [17], the value of mean shear stress along the shear plane across the direction, *S =* 112.1 MPa, is similar to obtained results. Shear force indicates the force required to shear the workpiece material along the shear plane in perpendicular machining. According to the experiment results, with an increase in *d*, there is an increase in *S*, although a change in *γ* in the positive direction significantly reduces the value of *S*. The characteristics of *S* changes are reflected in the chip-formation characteristics, which are described in more detail in the next part of this paper. Both Liao et al. [15] and Sugita et al. [32] indicate that the formation of a continuous chip at a small depth of cut, and the formation of a broken chip for a large depth of cut, results from the shear force value. This assumption is reflected in the above research results and confirms the scientists’ conclusions. High values of shear force in the shear plane result in the discontinuity of the chip.

### 3.3. Specific Cutting Force

The specific cutting force was determined based on the value of the *F_c_* obtained experimentally. 

First, the obtained results allow us to conclude that at a depth of cut *d* = 10–50 μm, there is a transition from micro- to macro-cutting in line with the size effect. In the range of 0.5 to 25 μm, there is a non-linear decrease in the specific cutting force value, and the decrease is linearly continued for the value of 50 to 175 μm. The border is not permanent. The curves for *d* < 25 μm assume *k_c_* at the level of GPa. For *d* > 25 μm, the *k_c_* drops to the level of MPa. For *d* < 25 μm, there is no effect of *γ* and *α* on the value of *k_c_*. In the case of *d* > 25 μm, there is a noticeable decrease in the value of *k_c_* with a change of *γ* in the positive direction, with no influence of *α* (see Figure 10).

The results show a significant influence on the material’s microstructure resulting from the heterogeneity of the composite structure of bone tissue. First, the osteons’ orientation influences the value of *k_c_* (the highest values characterize the transverse direction). The cutting direction is the most regular. According to the size effects, the microstructure of the processed material influences the cutting mechanism [42]. The diameter of the osteons ranges from 150 to 350 µm, and each is surrounded by a 5 µm-thick cement line [43]. Osteons are separated by an interstitial matrix, assuming a thickness of 50 μm. The small size and irregularity of the arrangement may cause a marked increase in the *k_c_* value for *d* < 25 μm. The size effect is also related to the correlation of the blade corner radius with the material’s microstructure, which results in the definition of minimum chip thickness [44]. The cutting tool corner radius was *R =* 104 ± 12 μm. According to the assumptions, for the cutting process to occur, *d* must have a value in the range of 0.05 r to 0.38 r [45]. According to the measurement of *r*, *d_min_* ranges from 5.2 to 39.5 μm. Thus, the presented results are in line with the theoretical foundations. In micromachining, due to the assumption of the *d* value at the *d_min_* level, the shear stresses are distributed around the cutting edge, and the material is mainly deformed [46]. Above *d_min_*, conventional cutting occurs where shear occurs along the shear plane. Machining with *d* < *d_min_* results in an increase in slipping forces, contributing to an increase in cutting forces and an increase in surface roughness [47].

Liao et al. [15] obtained *k_c_* tests at the level of 100–400 MPa in the direction of parallel, 100–400 MPa in the direction of the across and 250–750 MPa in the direction of the transverse for *d* = 150-5 μm. Only one tool with the following parameters was introduced: *γ* = 8°, *α* = 8° and *R =* 1 μm. Bai et al. [16] do not provide the results of *k_c_*, but based on the results of the measurement, *F_c_* and *γ* = 10°, *α* = 7°, the width of cut 3 mm and *d* = 10, 50, 150 μm, *k_c_* can be developed. Comparing them with the tools used in the study—*γ* = 10°, *α* = 5° and *R =* 104 ± 12 μm—the results presented in Figure 11 were obtained. It should be noted that Liao et al. [15] used a cutting speed of 33 mm/min, while the author used 30 mm/min. Bai et al. [16] did not present the value of *r*. The results indicate the similarity of the obtained analyses, but a significant influence of *R* on *k_c_* should be noted.

According to the observations and the comparison, the bone-cutting mechanism is influenced by the anisotropy of the material and the machining parameters. In a direction parallel to the osteons, the lowest level of specific cutting force is required (see Figure 10). The noticeable increase in *k_c_* results from the phenomenon of minimal uncut chip thickness, which is related to *r*. In the conducted studies, r was significantly higher than in Lia et al. [15]; therefore, the similarity of the results was observed for *d* > 25 μm.

### 3.4. Coefficient of Friction (COF)

The value of the COF should be divided into several ranges according to the cut depth of cut *d*_min_ resulting from the minimal uncut chip thickness of *d_min_* = 10 μm and the *γ* value, which causes a change in the cutting process. In all cases, the highest values of *COF* are obtained for *γ* > 0° Λ *d* > 10, and the lowest for *γ* > 0° Λ *d* ≤ 10. It should be indicated that the factor determining the *COF* value is *d*. The slightest *COF* deviation characterizes the parallel direction, especially for *d* ≤ 10 compared to transverse and across runs (see Table 3). It should be emphasized to reduce data discrepancies for greater depths, which indicates the process’s stability and less susceptibility to interference. The value of the COF for negative rake angle is characterized by less dynamism of changes, which is a direct result of the cutting process.

Figure 12 show the changes in the *COF* curves for two groups *d* and a specific range of *γ* for the parallel direction and *α* = 5°. There is a noticeable tendency for a slight increase in *COF* with an increase in y for the *d* < 10 groups, and for the group *d >* 10 this course is the opposite. It should also be noted that for *γ >* 0, *COF* has similar values for selected *d*, and in the case of *γ* ≤ 0, these values have more significant discrepancies.

The average COFs are relatively consistent with the results obtained by other authors. It should be noted that no available results directly related to the tool–chip COF are available. Zhang et al. [48] designated the value of the COF for the surface-flattened cortical bone as 0.58 ± 0.06. The study included plane-to-plane friction tests. Yu et al. [49] resulted from tangential fretting tests on the cortical bone using titanium samples and obtained maximum *COF* values of 0.51. Similar results were obtained by Beimond et al. [50] by determining the average COF as 0.64 ± 0.04. The comparison indicates the correctness of the research but suggests the need to conduct further detailed analyses.

### 3.5. Chip-Formation Mechanism

Acoustic signal detection can be used to characterize the chip-formation process. Scientific research by numerous researchers, including Hase et al. [51], indicates a clear difference in the acoustic signal between the continuous-type chips and the serrated chips. According to Kakade et al. [52], a longer *AE* signal rise time indicates continuous chip formation. Barry and Brayne [53] concluded that *AE* is correlated with the plastic deformation work rate and sliding friction. Figure 13 shows an exemplary measurement result for the cutting force *F_c_* and the *AE* signal. Three runs were distinguished. Waveform I (black) occurs when the maximum chip-breaking force *F_c_* is reached, and the *F_c_* value drops completely. The *AE* signal increases and then decreases to *AE* = 0. This course occurs when elementary chips are formed. Course II (green) occurs as the *F_c_* drops to an *F_c_* value > 0, with a complete decline in *AE.* This process occurs in the case of arc, spiral short and spiral conical chip breaking. These are short chips of a repetitive shape. Chip formation is continuous, although the chips break. Course III (purple) is characterized by continuous spiral flat chips. The force *F_c_* builds up gradually and slowly, and the *AE* signal shows numerous crosswise cracks in the chip. An *AE* < 1 value also produces needles, powder and short chips.

This topic was extensively discussed by Bai [16], Sugita [19] and Liao [15], although they did not include negative rake angles in the research. The following results are a complement and a partial confirmation of their results.

#### 3.5.1. Across Direction

For depths above 150 μm, there was a fragmentary structure cracking with a tendency to delve into the material. Positive rake angles caused the formation of larger cracks, and negative angles caused severe pressure, with the phenomenon of chip extraction. Chips were freely formed for a depth of 125–10 μm for both negative and positive angles, generating cracks between osteons. Spiral or arc chips were formed. Smaller cutting depths resulted in similar effects as in the case of parallel movement. For positive angles, protruding surface irregularities were sheared, and for negative angles, the material was dented, and chips were formed in dust, needle or strips. The influence of the value of the angle of application of the *α* on the cutting mechanism was not noticed. Figure 14C is presented to compare the values of the recorded signals of the AE sensor regarding the value of signals, depending on the orientation of osteons during cutting with a tool with a rake angle *γ* = 20° at a depth of *d* = 25 μm. In the transverse direction, the highest maximum values of *AE_max_transverse_ =* 2.2 V were obtained, and for the other directions, *AE_max_parallel_ =* 0.35 V and *AE_max_across_* = 0.54 V. However, it should be pointed out that the intensity of obtaining AE values above the mean is the highest for across movement (553 occurrences), followed by that for parallel (245 occurrences) and transverse (80 occurrences). It primarily indicates the nature of chip formation. The transverse direction is uncontrolled, sudden and intense, while for other directions, there is regularization, with the greatest stabilization for the parallel direction. The average values of *AE* take *AE_avg_parallel_*
_=_ 0.01 ± 0.002 V and *AE_avg_across_* = 0.03 ± 0.002 V and *AE_avg_transverse_* = 0.04 ± 0.002 V (see Figure 14).

#### 3.5.2. Parallel Direction

For all the investigated rake angles for depths above 175 μm, elemental chips were formed in the form of tears, extracting fragments from the tissue structure. The tool sinks spontaneously into the tissue. For a depth of 150 μm to 5 μm, the chip-formation process occurred freely, without interruptions and increases. There was a noticeable tendency to delve deeper into the material for negative rake angles. The formed chips take the shape of a spiral. For values below 5 μm, tools with a positive rake angle scraped off small tissue fragments, levelling the surfaces. Tools with negative rake angles slipped on the surface, deforming the tissue plastically without cutting. Figure 15B presented a graph showing cracks, depending on the applied rake angle *γ*, for a cutting depth of 50 μm. There is a noticeable presence of fewer cracks for *γ* = 0°. The average values of the AE coefficient took the following values: *A _avg_40_* = 0.06 ± 0.02 V, *A _avg_0_* = 0.03 ± 0.01 V and *AE_avg_−40_* = 0.05 ± 0.01 V. The average values are similar, although they indicate the occurrence of more cracks for the cutting angle *γ* = 40°, and a decrease in the number of cracks towards the zero cutting angle and then the adhesion towards the negative direction. The maximum values obtained are: *AE_max_40_* = 3.15 V, *AE_max_0_* = 1.28 V and *AE_max_-−40_* = 2.56 V (see Figure 15). For the indicated depth of cut, the process is the most visible, and the change in depth in the growth direction causes an increase in the occurrence of cracks and in the decreasing direction results in the disappearance of the recording of AE regardless of the value of the rake angle *γ*.

#### 3.5.3. Transverse Direction

In the case of depths above 125 μm and positive rake angles, fragmentary chips formed due to cracking of the bone structure along the feed direction, perpendicular to the axis of osteons (Figure 14A). The cracks decomposed irregularly, and as the depth increased, reduced repeatability of formation was noted. However, there was no clear correlation between osteon size and chip size, as in the Bai et al. study [16]. In the case of negative rake angles, smaller elementary chips were formed and crushed during formation. As shown in the analysis of the fracture mechanism, chips were formed due to transverse cracking and pressure in the longitudinal direction. In both cases, it is necessary to emphasize the negative effect in the form of the formation of an irregular surface of bone tissue after the passage of the tool. The tool delved into the tissue, and then there was a fracture, detachment of a tissue fragment and repetition of the process. It was not a smooth phenomenon but rather an increasing one.

For depths from 10 μm to 100 μm and positive rake angles, the chips formed into a spiral shape of variable thickness were obtained. A similarity was noted in numerous cracks (crack spacing) along the edge of the chip, with a spacing of about 100 μm. As the depth decreased, longer chips were formed, and there were no cracks along the cutting edge. In the case of negative angles, stripe and needle chips were formed, and the material was pushed and crushed. In cases of the depth of 0.5 μm to 5 μm, a partial “slipping” of the tool on the surface was observed due to plastic deformation of cortical bone tissue. No cracks or marked changes were observed. The formed chips took the shape of a needle, strips or dust. For the zero rake angle, three processes should be distinguished: formation in the form of structure fracture (for depths above 100 μm) resulting in elementary chips, formation of a spiral chip with stable shaping (for 10 to 100 μm) and the formation of dust, needle chips without clear changes in the morphology of the surface. 

As indicated in Figure 16A, increasing the cut depth intensifies the cracks’ size. For *d* = 175 μm, the voltage value obtained by means of the AE sensor obtains maximums at the following levels: *AE_max_175_* = 6.13 V, *AE_max_50_* = 3.15 V and *AE_max_5_* = 0.03 V. On the indicated graph, the results for a depth of cut of 5 μm are imperceptible due to the absence of recorded cracks. The average AEavg values for individual depths of cut take the following: *AE_avg_175_* = 0.4 ± 0.1 V, *AE_avg_50_* = 0.06 ± 0.02 V and *AE_avg_5_* = 0.02 ± 0.01 V. The presented results indicate separate cutting processes, tissue cracking and chip formation. For a depth of 175 μm, significant cracks occur during chip formation. The process stabilizes for a depth of 50 μm, decreasing the cracks’ frequency and strength. For values below 5 μm, the process stabilizes at a deficient level, and no cracks have been recorded.

Among the three cutting directions, it should be emphasized that the transverse direction resulted in a much greater number of uncontrolled behaviors during chip formation. Vertical shear of osteons requires higher cutting force, and the morphology of the structure causes a cracking mechanism that is difficult to predict. Obsessiveness follows that the most predictable direction of movement is parallel movement. Referring to Liao et al. [15], a two-stage chip-formation process should be distinguished: I—plastic deformation and crack growth in the first stages of delving tool chip in cortical bone structure and initialization and crack propagation after energy accumulation in the shear zone. For *γ* > 0 it can be assumed that the energy release rate is proportional to the cutting depth, and the fracture occurs when it exceeds the critical value for *γ* < 0, the plastic deformation process is prolonged and crack growth is disturbed due to the impact of the cutting surface. According to the Ernst-Merchant model, *γ* affects the energy contribution to the shear crack. The amount of energy accumulated in the first stage is much higher than the positive rake angle, resulting in the propagation of stronger cracks deep into the bone tissue structure.

### 3.6. Chip Morphology

Figure 17 shows chip types obtained for the *γ* = 30° and clearance angle *α* = 15 °. The movement took place in a direction parallel to the arrangement of osteons. Starting from a depth of 200 μm, chips were shelled, tissue fragments chiseled with a heterogeneous structure, and for *d* = 175 μm fragmentary chips and short spiral chips with intensely cracked edges from the resulting stresses obtained. The length of the chip can be estimated as 1–3 mm, which may be related to the osteon size. Chips of a spiral nature were obtained from *d* = 150 μm to *d* = 1 μm. For *d* = 150 μm, the chips increased the number of turns, increased the diameter and reduced edge cracks. For *d* = 50 μm, the most pronounced chips with a uniform structure, without transverse cracks, were obtained. Reducing the depth of cutting resulted in increased chip irregularities, increased spacing between successive layers and numerous holes and transverse cracks. Below *d* = 5 μm, the chips disintegrate by cracking, picking and crushing. The spiral chips are already flattened and short for *d* < 2 μm. They are accompanied by the formation of needle chips and fine powder. Below *d* = 5 μm, a small number of chips take a spiral shape.

Table 4 shows a generalized description of chip shapes relative to depth *d* and rake angle *γ*. Three groups of chips should be distinguished: fragmentary, resulting from non-correlated cracks; spiral, where the forming was controlled and predictable; and dust-powder, where the tool slipped on the tissue surface, and abrasion of the top layer. Chips from the spiral group characterizing controlled machining occurred mainly in cutting depth range *d* = 150 to 2.5 μm and rake angle *γ* = 40–10°; above *d* = 150 μm elemental chips, needles were obtained, while below *d* = 2.5 μm mainly small needle chips and dust-powder were formed. For negative angles of the cutting, the tendency to form short, elemental needle chips prevailed, with a slight dominance of spiral chips for *d* = 150 to 5 μm and rake angle *γ* = −10°, and *d* = 10 to 5 μm and rake angle *γ* = −20°. For rake angle *γ* = 0°, quick results were obtained, and a precise cross-section of all the chips was formed.

Machining with large negative rake angles, the tool tends to plough the workpiece material instead of forming chips and demonstrates the typical topography of burrs. These observations of Akbari et al. [41] confirm the results of the above studies.

It should be noted that the determined minimal uncut chip thickness at 5.2 to 39.5 μm is reflected in the chip-formation mechanism and chip morphology. For depths below 50 μm, the presence of short, fragmentary chips with a tendency to form a spiral, needle chips and finally powder with a decrease in *d*, is indicated. Similarly, the analyzed *AE* signals show a noticeable decrease in cracking and sack formation below these values.

## 4. Conclusions

This study performed experiments on the orthogonal cutting of the cortical bone tissue. A wide range of cutting tool geometries and cut depths was used to understand the bone-removal mechanism better. The cutting-edge geometry (rake angle *γ*) has influenced the cutting process. The effect of cortical bone anisotropy and cutting depth *d* on chip formation, cutting force and coefficient of friction (*COF*) was also analyzed. The following conclusions can be drawn from the experimental results and analyses:The value of the resultant cutting force *R* is most influenced by the osteons’ orientation, the rake angle *γ* and the depth of cut *d*:an increase in the depth of cut *d* causes an increase in *R*;for the transverse direction, the highest *R* values were recorded;transition rake angle *γ* in the negative direction increases *R*.

The most optimal parameters for cutting, assuming the goal of lowest resultant cutting force *R* results, are: *γ* = 10°, *d* < 50 μm and parallel cutting direction.

2.The rake angle *γ* most influences the shear force value *S*. It decreases linearly with the increase of the rake angle *γ* in the positive direction. The highest *S_max_* values are recorded for the transverse direction.3.The rake angle *γ* does not affect the specific cutting force *k_c_* value for *d* ≤ 25 μm. For *d* > 25 µm, there is an increase in *k_c_* with a change in *γ* in the negative direction. The value of the cutting edge radius *r* significantly influences *k_c_*.4.*COF* depends on the depth of cut *d* and the rake angle *γ*. Two groups were distinguished:for *d* > 25 μm, the *COF* value decreases with the increase of *γ*;for *d* < 25 μm, the *COF* value increases with an increase in *γ*.5.*AE* signals characterize the chip-formation process and distinguish cutting parameters such as direction, depth and rake angle.6.The chip-formation mechanism significantly influences the direction of the osteon’s orientation, the depth of cut *d* and the rake angle *γ*. The depth of cut has the most significant influence on chip morphology.7.The geometry of the cutting tool, and ultimately the abrasive grain, significantly impact the cutting process.

The novelty of research resulting from a wide range of cutting tool geometry confirms the possibility of using abrasive machining for cortical tissue machining. It also introduces new information on chip formation and process-related phenomena. The analysis showed that the use of cutting tools with cutting geometry comparable to abrasive grains can significantly affect the optimization of the process. It should be emphasized that a controlled cutting process takes place in the entire range of the depth and geometry of the grains. Despite the visible differences in the values of cutting forces and the type of chips formed, the use of negative rake angles allows for the machining of tissues. This study confirms that abrasive grain with undefined geometry could fulfill its function and be used in surgical procedures. In this case, further detailed studies of the treatment of the bone surface with abrasive machining should be carried out.

## Figures and Tables

**Figure 1 materials-15-06414-f001:**
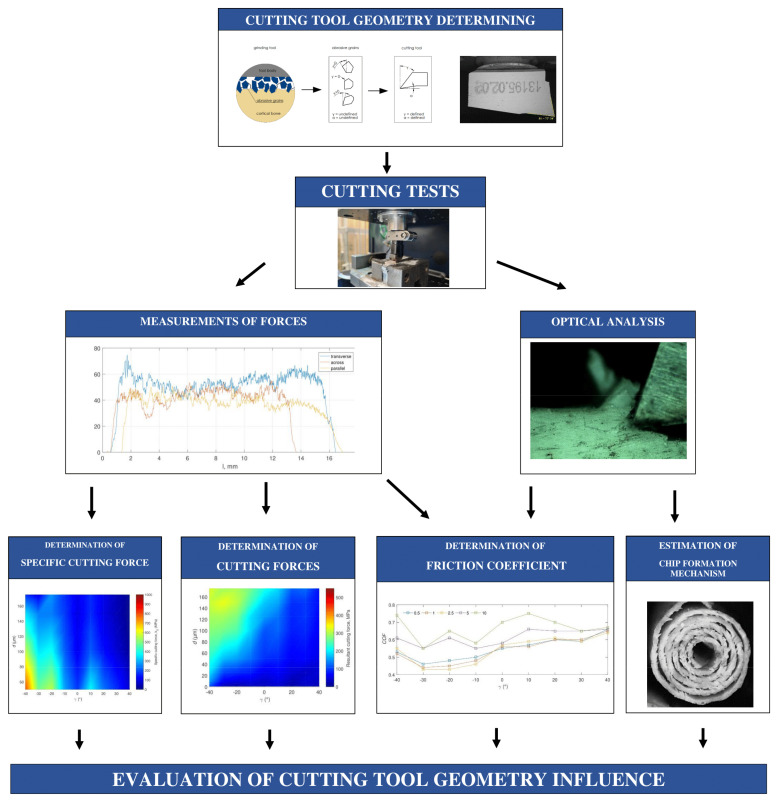
Scheme of conducted experiments.

**Figure 2 materials-15-06414-f002:**
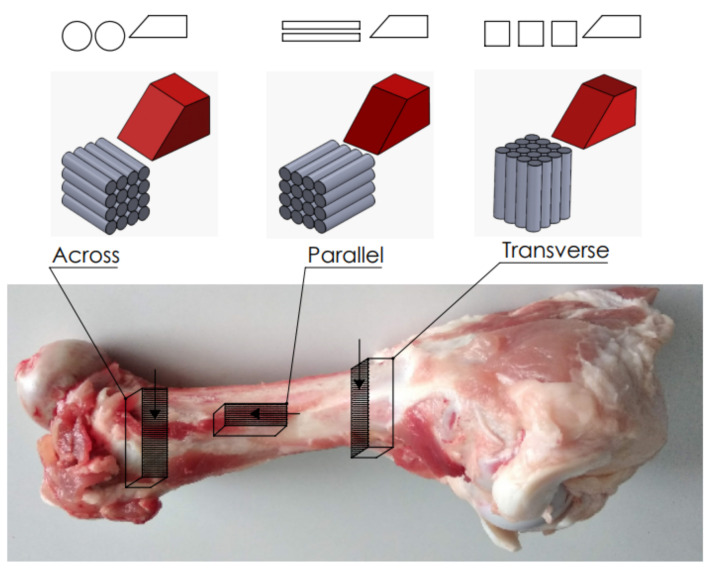
Preparation of bone specimens for transverse, across and parallel directions of osteons orientation.

**Figure 3 materials-15-06414-f003:**
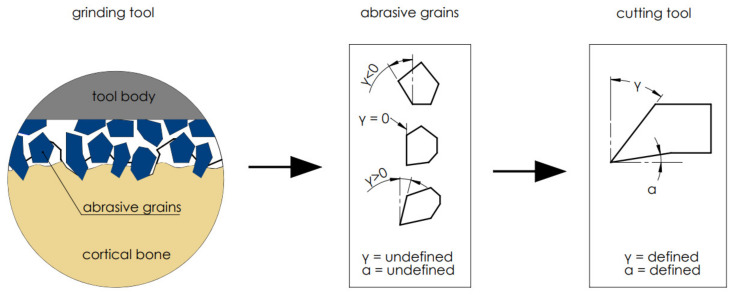
Diagram of simplifying cutting geometry with an abrasive tool. Transfer of cutting from abrasive grains with undefined geometry to cutting tools with a defined geometry.

**Figure 4 materials-15-06414-f004:**
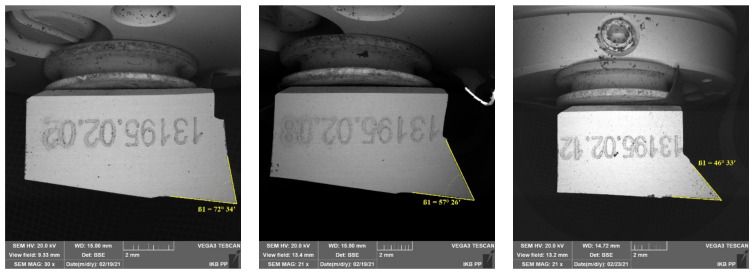
SEM photos of the cutting edges, showing the defined cutting edge geometry.

**Figure 5 materials-15-06414-f005:**
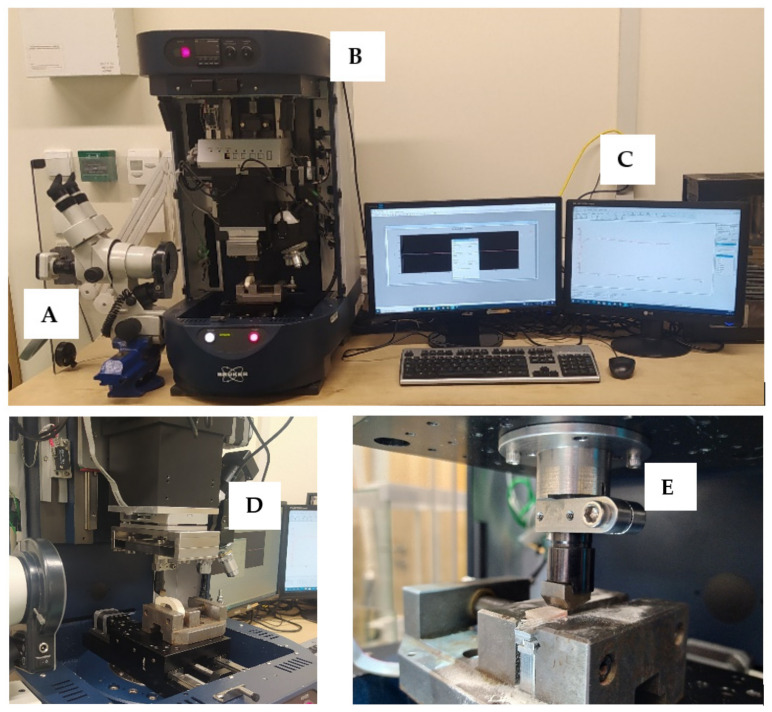
Cutting process parameters measurement station: (**A**) optical microscope with a camera, (**B**) Tribotester Bruker UMT, (**C**) control and data recording system, (**D**) force sensor DFM-20, (**E**) cutting tool holder.

**Figure 6 materials-15-06414-f006:**
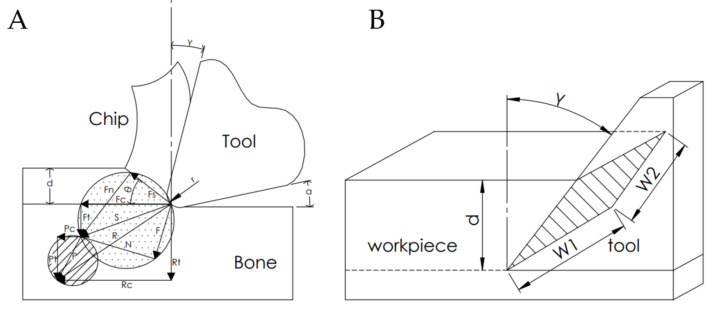
Schematic diagram of (**A**) the used cutting model, and (**B**) the contact zone of the workpiece and cutting tool.

**Figure 7 materials-15-06414-f007:**
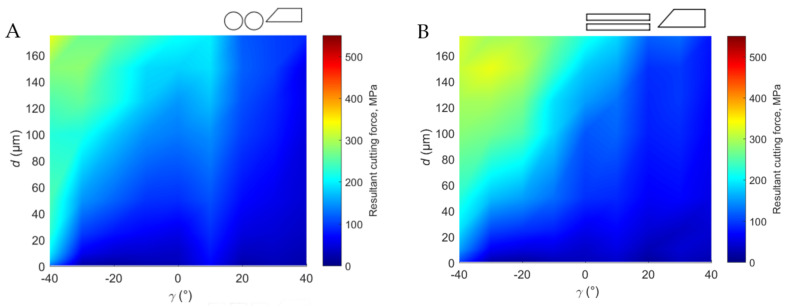
Charts of values of resultant cutting force depending on the rake angle and depth of cut in specific directions of tool movement: (**A**)—across, (**B**)—parallel, (**C**)—transverse and (**D**)—a resultant cutting force for transverse cutting direction (2D graph). The cutting directions are shown in Figure 2.

**Figure 8 materials-15-06414-f008:**
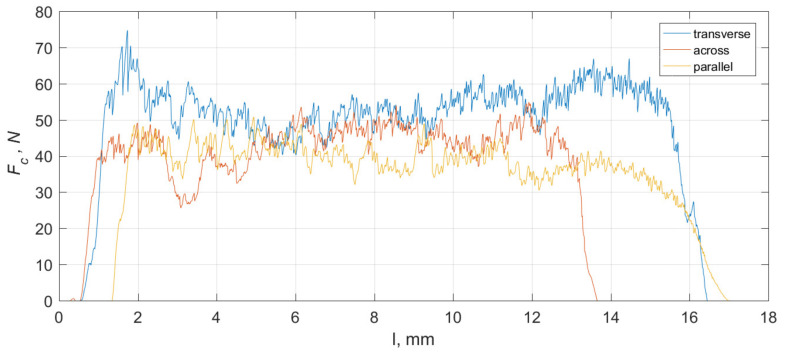
The curves of change values of *F_c_* for three cutting directions: transverse, across and parallel, for *γ* = 30°, *α* = 5° and *d* = 25 μm.

**Figure 9 materials-15-06414-f009:**
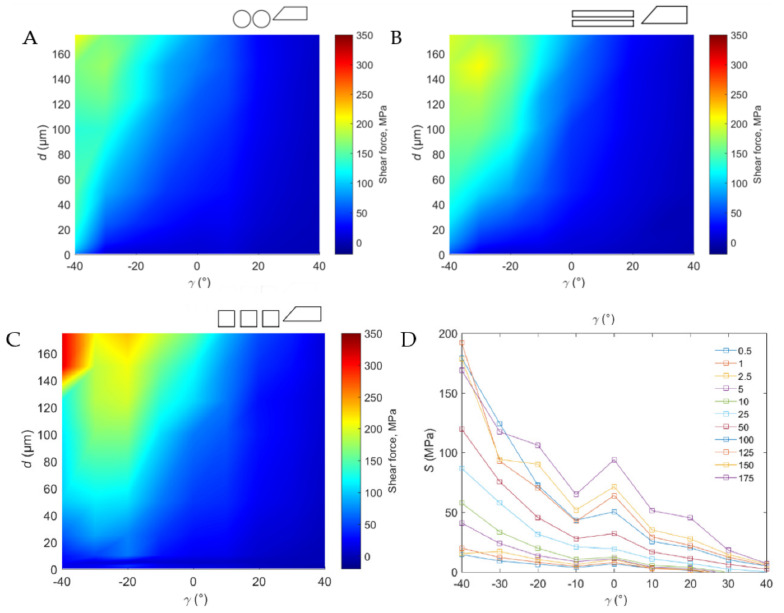
Shear force depending on the rake angle and depth of cut in specific directions of tool movement: (**A**)—across, (**B**)—parallel, (**C**)—transverse and (**D**)—resultant cutting force for transverse cutting direction (2D graph). The cutting directions are shown in Figure 2.

**Figure 10 materials-15-06414-f010:**
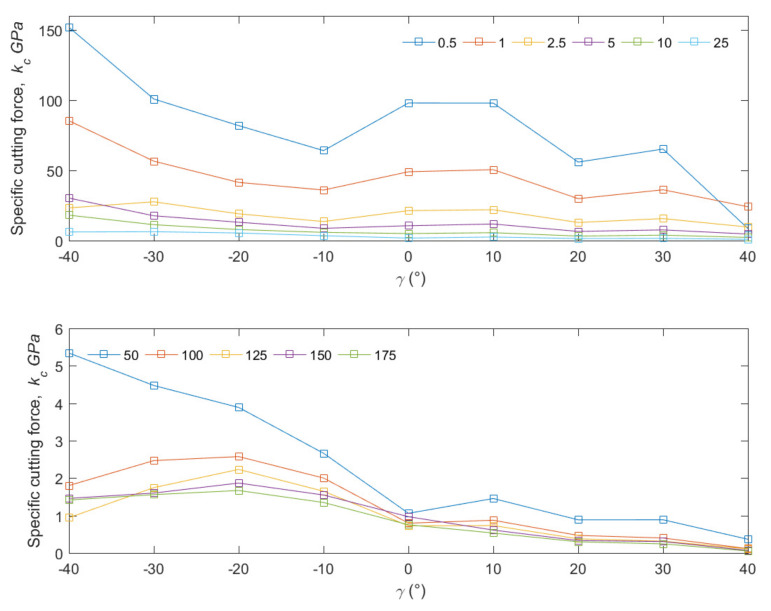
The curves of changes of specific cutting force for across cutting direction, *α* = 5°, and different depths of cut.

**Figure 11 materials-15-06414-f011:**
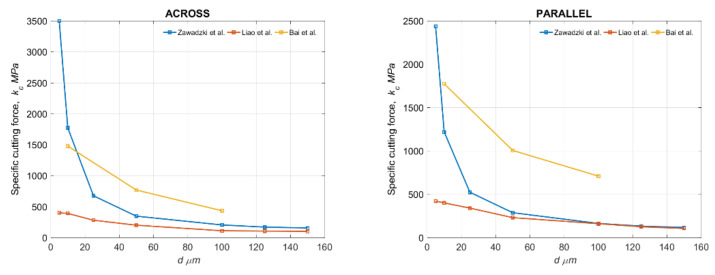
Comparison of values of specific cutting force for three cutting conditions according to the results obtained by Liao et al. [15], Bai et al. [16] and current results.

**Figure 12 materials-15-06414-f012:**
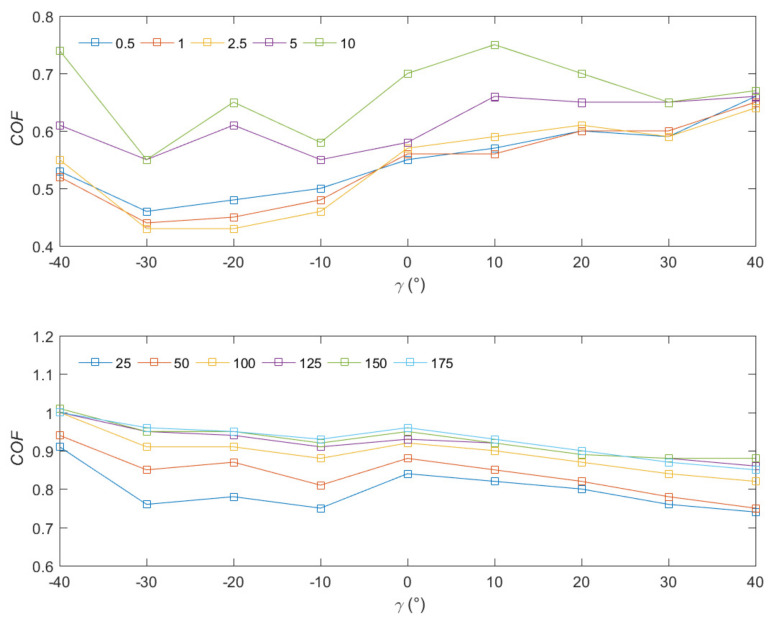
Comparison of the COF for the orthogonal cutting in the parallel direction and *α* = 10°.

**Figure 13 materials-15-06414-f013:**
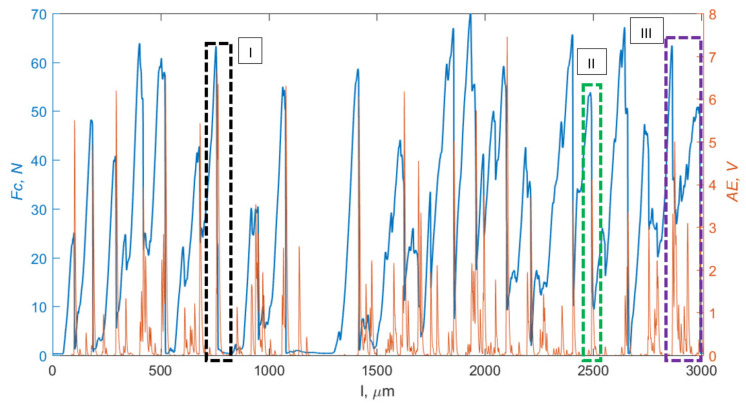
Comparison of *F_c_* and *AE* courses as a function of path *l*, with three characteristic courses for forming: I—elementary chips; II—arc, spiral short and spiral conical chips; III—continuous chips.

**Figure 14 materials-15-06414-f014:**
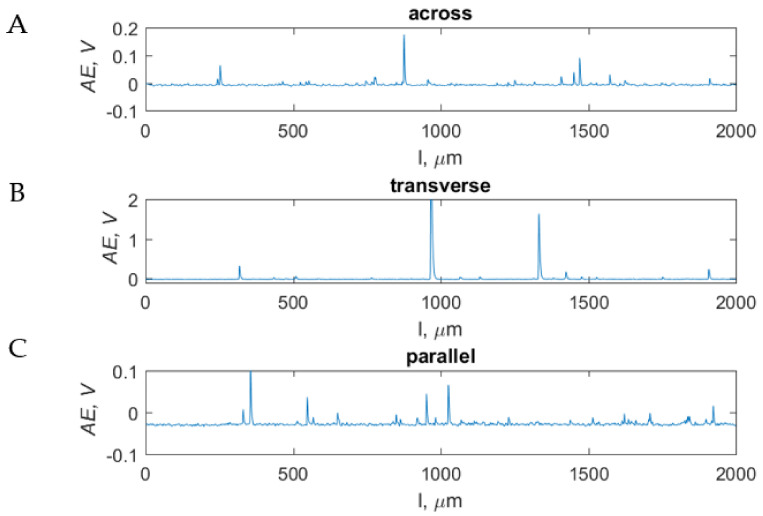
Measurement results of the AE sensor for chip-formation processes during orthogonal cutting at a constant *d* = 25 μm and *γ* = 20°, in the across (**A**), transverse (**B**) and parallel (**C**) directions.

**Figure 15 materials-15-06414-f015:**
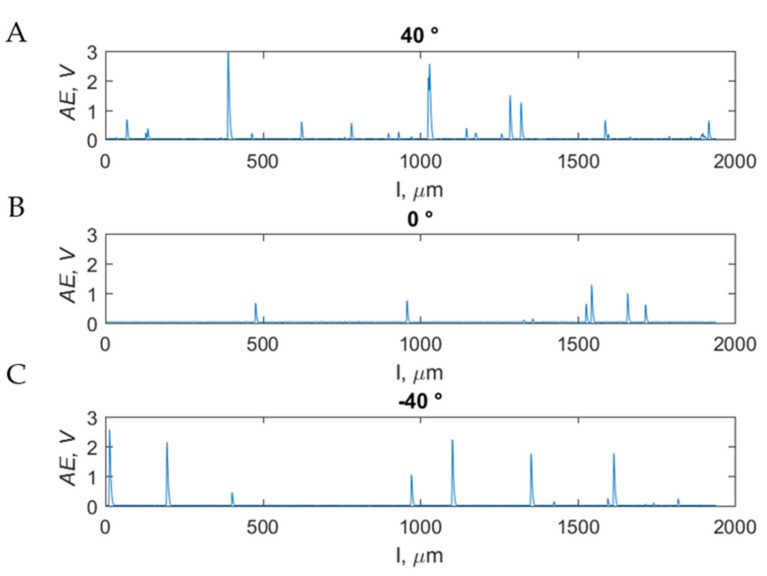
Measurement results of the AE sensor for chip-formation processes during orthogonal cutting at a constant *d* = 50 μm in the parallel direction for different rake angles: (**A**) *γ* = 40°, (**B**) *γ* = 0° and (**C**) *γ* = − 40 °.

**Figure 16 materials-15-06414-f016:**
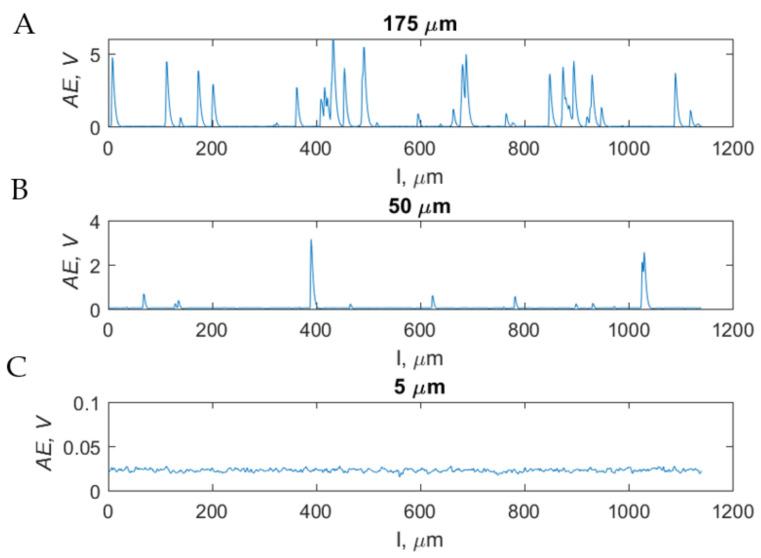
Measurement results of the *AE* sensor for chip-formation processes during orthogonal cutting: at a constant rake angle *γ* = 40° in the perpendicular direction with a constant cutting depth (**A**) *d* = 175 μm, (**B**) *d* = 50 μm and (**C**) *d* = 5 μm.

**Figure 17 materials-15-06414-f017:**
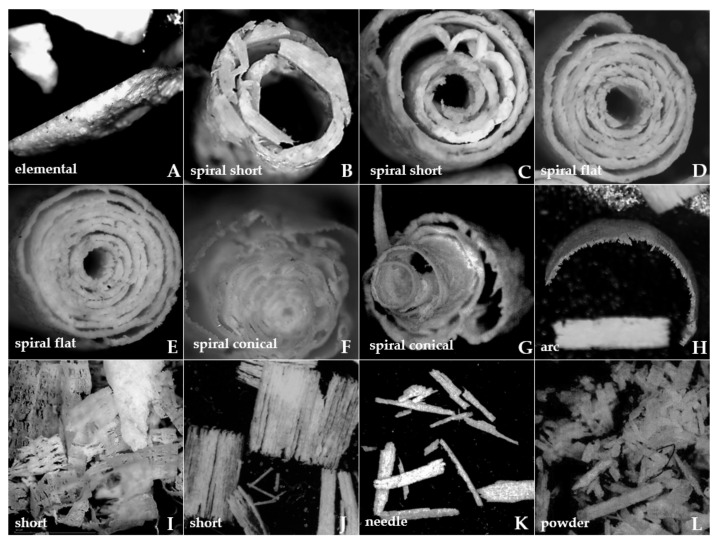
View of chips resulting from machining with *γ* = 30° and *α* = 15° in the parallel direction for the following depths *d* of cutting: (**A**) 200 μm, (**B**) 175 μm, (**C**) 150 μm, (**D**) 125 μm, (**E**) 100 μm, (**F**) 50 μm, (**G**) 25 μm, (**H**) 10 μm, (**I**) 5 μm, (**J**) 2,5 μm, (**K**) 1 μm, (**L**) 0.5 μm.

**Table 1 materials-15-06414-t001:** Material properties of cortical bone were assumed for carrying out the research.

Parameter	Value
Elastic modulus (GPa)	20.4
Shear (GPa)	4.14
Poisson’s Ratio	0.325
Strength (MPa)	134
Elastic energy release ratio (J/m^2^)	860
Density (kg/m^3^)	2000

**Table 2 materials-15-06414-t002:** Input parameters and test setup configuration method.

Number	Parameter	Unit	Value
I	rake angle	*γ*, °	40, 30, 20, 10, 0, −10, −20, −30, −40
II	clearance angle	*α*, °	5, 10, 15
III	cutting depth	*d*_,_ μm	0.5, 1, 2, 2.5, 5, 10, 25, 50, 100, 125, 150, 175
IV	cutting velocity	*v_c_*_,_ mm/min	30
V	cutting direction	details in Figure 2	across, parallel, transverse
**Parameter configuration scheme ***			
I + II + III + IV + V → Results			
**Configuration example **			
−40° + 15° + 175 μm + 30 mm/s + Transverse → *F_c_*, *F_x_*, *AE*, Chip morphology			

* 972 measurements were made.

**Table 3 materials-15-06414-t003:** Results of measurements of the value of the COF.

Direction	*α*					
	5°		10°		15°	
Transverse	COF	δ	COF	δ	COF	δ
*γ* > 0° Λ *d* ≤ 10	0.53	±0.19	0.53	±0.14	0.67	±0.09
*γ* > 0° Λ *d* > 10	0.92	±0.09	0.90	±0.10	0.93	±0.05
*γ* ≤ 0° Λ *d* ≤ 10	0.61	±0.09	0.68	±0.10	0.63	±0.04
*γ* ≤ 0° Λ *d* > 10	0.85	±0.06	0.85	±0.08	0.85	±0.06
Parallel						
*γ* > 0° Λ *d* ≤ 10	0.53	±0.08	0.53	±0.08	0.69	±0.10
*γ* > 0° Λ *d* > 10	0.88	±0.09	0.91	±0.07	0.95	±0.06
*γ* ≤ 0° Λ *d* ≤ 10	0.62	±0.05	0.62	±0.04	0.79	±0.12
*γ* ≤ 0° Λ *d* > 10	0.83	±0.06	0.86	±0.06	0.86	±0.06
Across						
*γ* > 0° Λ *d* ≤ 10	0.62	±0.17	0.58	±0.11	0.51	±0.13
*γ* > 0° Λ *d* > 10	0.95	±0.05	0.92	±0.06	0.94	±0.05
*γ* ≤ 0° Λ *d* ≤ 10	0.67	±0.16	0.69	±0.11	0.58	±0.10
*γ* ≤ 0° Λ *d* > 10	0.87	±0.08	0.88	±0.07	0.87	±0.06

**Table 4 materials-15-06414-t004:** Characteristics of the geometry of chips formed during the test depending on rake angle *α* and depth of cut *d*. The letters in the table refer to the chip photos shown in Figure 16.

*γ*°	*d* μm											
	200	175	150	125	100	50	25	10	5	2.5	1	0.5
40	A	B, C, A	B, C	B, C	D, E	D, E	D, E	I, J, H	K	K	L	L
30	A	B, C	B, C	D, E	D, E	F, G	F, G	H	I, J	I, J	K	L
20	A	B, C	B, C	D, E	D, E	F, G	F, G	I, J, H	I, J	I, J	K	L
10	A	B, C	B, C	D, E	D, E	F, G	F, G	I, J, H	I, J	I, J	K	L
0	A	H	H	H	H	K	K	K	K	K	L	L
−10	A	A	B, C, J	B, C	B, C	B, C	B, C	I, J	I, J	K	K	L
−20	A	A, K	A, K	A, K	H	H	B, C	I, J	I, J	K	K	L
−30	A	A, K	A, K	A, K	A, K	A, K	A, K	A, K	A, K	K	K	L
−40	A	A, K	K	K	K	K	K	K	K	K	K	L

Legend: A—elemental, B and C—spiral short, D and E—spiral flat, F and G—spiral conical, H—arc, I and J—short, K—needle, L—powder.

## Data Availability

Not applicable.

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
