# Peer review of "Influence of Machining Parameters on Cutting and Chip-Formation Process during Cortical Bone Orthogonal Machining"

_materials, 2022, doi:10.3390/ma15186414_

Round 1
Reviewer 1 Report
This paper focuses on the bone cutting process and the influence of tool geometry and depth of cut on the force, chip morphology, fiction coefficient, and AE. The novelty of this paper should be further emphasized, and the language should be critically improved. Other comments are as follows.
1. The abstract needs a critical improvement and addition of quantitative results. The results showing in the abstract is poor, since the influence of the grain geometry on the cutting and chip formation is not even included, which based on the authors is the goal of this paper.
1. Please give a detailed explanation on the “ grain of bones” and “grain geometry of bones”, the current form is quite confusing to the readers.
2. Nomenclature is needed.
3. Please check the spell and grammar carefully, and uniform the symbols and variables.
4. The directions introduced in section 3.1 should have a illustration, e.g., which is the across direction, what is the transverse direction?
Line 455, what does the symbol “Λ” mean ?
5. What is the function of acoustic emission? How will the AE signal reflect the chip formation process? How do the authors find the crack or chip formation by the AE. In addition, what is lm in Fig. 12?
6. Please improve the conclusion part, the current form is so difficult to read and understand, and the units are needed to add, and what do the variables e.g., S, d, d, y, kc, kc, w, etc., stand for, especially with the same spell but in different format (such as kc and kc, are they the same or not?)
7. The figures for different chip geometries should be given in Table 4, such as the figures for “elemental needle” or “needle”. Or the chip geometries should be relative with chip morphology showing in Fig. 15.
Author Response
Dear Reviewer,
Thank you very much for the opportunity to correct the manuscript. Following the guidelines presented above, I made changes throughout the document. Additionally, referring to the indicated amendments:
- The abstract has been modified to reflect better the research results and the novelty of the proposed machining method.
- A general nomenclature was introduced for the entire text (line 104).
- Symbols and markings have been rechecked and corrected.
- Cutting directions have been added to Figures 6, 8, and 10 for clarity (lines 215, 380 and 447)
- The symbol "and" in Table 3 (line 471) has been corrected. The wording "γ> 0 ° ∧ d ≤ 10" means γ> 0 ° "and" d ≤ 10.
- The acoustic emission helps characterize the ways that a chip is formed. Figure 12 (line 504) has been added to clarify the analysis of the AE signal. Additional literature items were also introduced, confirming the AE analysis's correctness.
- The "Conclusions" section has been completely redrafted and changed. Readability and marking of individual parameters have been improved, and the novelty of the research was also emphasized.
- Figure 16 and Table 4 have been linked using similar notations. This increased the readability of the entire subsection.
- I did not find phrases like "grain of bones" and "grain geometry of bones" in the text. However, the text was revised and checked to avoid this wording.
- The linguistic and stylistic correction was also made. The author would like to point out that if the language of the article still requires corrections, he will direct him to English language editing arranged by MDPI.
Additionally, following the comments of other reviewers, the following changes were introduced:
- The article's title has been changed to reflect the work's content and research.
- The plan of the experiment has been extended. A parameter configuration diagram has been added in Table 2 (line 188).
I hope I met the suggested requirements. I am ready to introduce further amendments.
Yours faithfully,
Paweł Zawadzki
Reviewer 2 Report
- Title is misleading. The cutting tool appears like a single point cutting tool but the title says abrasive machining. Please provide clarity on this.
- Quantified results need to be presented in the Abstract.
- Minor language and grammatical editing are needed.
- Plan of experiments needs to be provided in a tabular manner listing the different process parameter variations carried out during the experiments.
- Increase the font size of text in the figures to make them legible.
Author Response
Dear Reviewer,
Thank you very much for the review of the article and the possibility of making corrections. Below I provide information on the changes made to the comments:
- The article's title has been changed to better reflect the work's content and research.
- The abstract has been modified to reflect better the research results and the novelty of the proposed machining method.
- The plan of the experiment has been extended. A parameter configuration diagram has been added in Table 2 (line 188).
- The font size in charts has been increased to improve readability.
- The linguistic and stylistic correction was also made. The author would like to point out that if the language of the article still requires corrections, he will direct him to English language editing arranged by MDPI.
Additionally, following the comments of other reviewers, the following changes were introduced:
- A general nomenclature was introduced for the entire text (line 104).
- Symbols and markings have been rechecked and corrected.
- Cutting directions have been added to Figures 6, 8, and 10 for clarity (lines 215, 380 and 447)
- The symbol "and" in Table 3 (line 471) has been corrected. The wording "γ> 0 ° ∧ d ≤ 10" means γ> 0 ° "and" d ≤ 10.
- The acoustic emission helps characterize the ways that a chip is formed. Figure 12 (line 504) has been added to clarify the analysis of the AE signal. Additional literature items were also introduced, confirming the correctness of the AE analysis.
- The "Conclusions" section has been completely redrafted and changed. Readability and marking of individual parameters have been improved, and the novelty of the research was also emphasized.
- Figure 16 and Table 4 have been linked using similar notations. This increased the readability of the entire subsection.
I hope I met the suggested requirements. I am ready to introduce further amendments.
Yours faithfully,
Paweł Zawadzki